# Assessment of Genetic Diversity of the Salangid, *Neosalanx taihuensis*, Based on the Mitochondrial COI Gene in Different Chinese River Basins

**DOI:** 10.3390/biology11070968

**Published:** 2022-06-27

**Authors:** Di-An Fang, Miao He, Ya-Fei Ren, Hui Luo, Yan-Feng Zhou, Shu-Lun Jiang, Yang You

**Affiliations:** 1Key Laboratory of Freshwater Fisheries and Germplasm Resources Utilization, Ministry of Agriculture and Rural Affairs, Freshwater Fisheries Research Center, Chinese Academy of Fishery Sciences, Wuxi 214081, China; ryff6601@163.com (Y.-F.R.); zhouyf@ffrc.cn (Y.-F.Z.); jiangshulun@ffrc.cn (S.-L.J.); 2National Demonstration Center for Experimental Fisheries Science Education, Shanghai Ocean University, Shanghai 201306, China; hemiao19971105@163.com (M.H.); lhlhlh0928@163.com (H.L.)

**Keywords:** aquatic ecology, diversity assessment, *Neosalanx taihuensis*, germplasm resources, population genetic structure

## Abstract

**Simple Summary:**

In the current study, we estimate the genetic diversity of the salangid *Neosalanx*
*taihuensis* sampled from 11 populations in the six typical river basins of China. Using the COI gene sequencing technology, the *N.* *taihuensis* population’s genetic difference within and between river basins was investigated. Significant levels of genetic subdivision were detected among populations within basins rather than between basins. Population history dynamics showed that *N.* *taihuensis* populations experienced a population expansion during the glacial period in the late Pleistocene. These results suggest that different populations should be considered as different management units to achieve effective conservation and management purposes.

**Abstract:**

The salangid *Neosalanx taihuensis* (Salangidae) is a commercially important economical fish endemic to China and restricted to large freshwater systems with a wide-ranging distribution. This fish species has continuous distribution ranges and a long-introduced aquaculture history in Chinese basins. However, the research on its population genetic differentiation within and between basins is very limited. In this regard, 197 individuals were sampled from 11 populations in the Nenjiang River Basin (A1–A4), Songhua River Basin (B1), Yellow River Basin (C1–C2), Yangtze River Basin (D1), Lanchang River Basin (E1–E2) and Huaihe River Basin (F1). Based on the COI sequence, the *N.*
*taihuensis* population’s genetic difference within and between river basins was investigated. The haplotypes and their frequency distributions were strongly skewed, with most haplotypes (*n* = 13) represented only in single samples each and thus restricted to a single population. The most common haplotype (H4, 67/197) was found in all individuals. The analysis of molecular variance (AMOVA) revealed a random pattern in the distribution of genetic diversity, which is inconsistent with contemporary hydrological structure. The mismatch between the distribution and neutrality tests supported the evidence of a population expansion, which occurred during the late Pleistocene (0.041–0.051 million years ago). Significant levels of genetic subdivision were detected among populations within basins rather than between the six basins. Population history dynamics showed that *N. taihuensis* experienced an expansion during the glacial period in the late Pleistocene. Therefore, different populations should be considered as different management units to achieve effective conservation and management purposes. These results have great significance for the evaluation and exploitation of the germplasm resources of *N. taihuensis*.

## 1. Introduction

The genetic structure of a population is dependent on the interaction of the species biology and the environment in which it resides. Populations from different basins often show significant genetic differences resulting from isolation, while a basin population demonstrates no or low levels of genetic difference [1]. There is evidence that geographic barriers can restrict gene flow and induce high levels of genetic differentiation between populations [2,3]. However, freshwater fish species populations from different basins often show a significant genetic differentiation resulting from geographic isolation [4].

The salangid, *Neosalanx taihuensis* (*Neosalanx tangkahkeii taihuensis* Chen, 1956) is a usually one-year-lived freshwater fish species from the Salangidae family, which is distributed mainly in the Yangtze River and affiliated lakes and which has a continuous distribution in Chinese river basins [5]. Because of its high commercial value, *N. taihuensis* has therefore been introduced to numerous lakes and reservoirs in more than 20 Chinese provinces since the 1980s [6]. It very rapidly adapted to the new conditions present in these lakes and reservoirs and quickly became the invaded fish species. As is known to all, different rivers have different habitat characteristics, and the fish adapted to them will also evolve different ecological adaptations, resulting in population differentiation or the formation of new species. In China, different river basins have a significant geographical isolation. The Yangtze River is the largest river in Asia and China, with more than 3000 tributaries and 4000 connected lakes [7,8]. The Lanchang River is one of the largest and most significant transboundary rivers in the world and is characterized by a high fish biodiversity [9,10]. Both basins are located in southern China; the Huaihe River is an important river in central China, with Yangtze River in the south and Yellow River in the north [11]. The Yellow River is the second longest river in China, which flows through one of the largest and thickest loess deposits in the world [12]. The Nenjiang River is one of the largest tributaries of the Songhua River, which is located in northeast China. There are many important wetland nature reserves, which sustain diverse ecosystems [13]. *N. taihuensis* has been studied for more than 100 years, but most of the studies have focused on taxonomy [14], biology [15] and molecular phylogeny [16]. However, there are few systematic studies on the biological adaptability and genetic germplasm resources of the transplanted *N. taihuensis.*

Molecular markers are useful tools for the assessment of genetic diversity and population structure, in addition to the various influential factors. It is well known that nuclear and mitochondrial (mtDNA) DNA markers are extensively used to evaluate genetic diversity and population structure in freshwater and marine fish species. MtDNA allows for the study of the geographic structure and historical demography of a species, because it is maternally inherited without recombination and has a relatively fast mutation rate [17]. It is currently understood that the genetic variation level influences the adaptability to changing conditions in a population. Therefore, in order to better understand the remarkable invasion effect by *N. taihuensis*, a status analysis of its genetic diversity is necessary.

The evolution speeds of mitochondrial cytochrome oxidase I (COI) is suitable for the detection of population differences and have been widely studied [18]. It is one of the most thoroughly studied mitochondrial genes and is ideal as a molecular marker [19]. In previous studies, COI barcode identification was able to solve the problem of species replacement in international trade [20]. Similarly, the genetic structure of Satoh’s analysis of 200 species of fish and its variation characteristics could explain the relationship between the species [21]; moreover, based on mtDNA COI, concatenated sequences identified a low genetic diversity and significant genetic differentiation in *Culter alburnus* from Chinese freshwater lakes [22]. In this regard, the COI gene sequence from 11 populations was analyzed in order to detect the population structure and the genetic diversity of N. taihuensis at large geographical scales. Samples were collected from the Nenjiang River Basin (NjRB, A1–A4), Songhua River Basin (ShRB, B1), Yellow River Basin (YeRB, C1–C2), Yangtze River Basin (YzRB, D1), Lanchang River Basin (LcRB, E1–E2) and Huaihe River Basin (HhRB, F1).

## 2. Materials and Methods

### 2.1. Sample Collection

In total, 197 individuals of *N. taihuensis* were collected from 11 populations in the NjRB (A1–A4), ShRB (B1), YeRB (C1–C2), YzRB (D1), LcRB (E1–E2) and HhRB (F1) from November to December in 2019 (Figure 1 and Table 1). To eliminate the possibility that these samples came from related individuals (schools of fishes), we selected many patches from one location and sampled them in triplicate for the same population. All individual tailfins were stored in 95% ethanol till DNA extraction. All fish experimental procedures were performed according to the Regulations for the Administration of Affairs Concerning Experimental Animals. Fish sampling were approved and authorized by the Fish Resource Committee in China.

### 2.2. DNA Extraction, PCR Amplification and Sequencing

Total DNA was extracted from the fish tailfins using E.Z.N.A.^®^ Tissue DNA Kit (Omega Biotek Store, Norcross, GA, USA) following the Kit protocol. The COI fragment was amplified using COIF: 5′-CCTTTGGCAGGAAACTTGGC-3′ and COIR: 5′-TGGTAAAGAATGGGGTCGCC-3′ as primers in a 50 μL reaction volume mixture containing 25 μL Premix Taq (1.25 U rTaq polymerase, 0.4 mM of each dNTP Mixture, 4 mM Mg^2+^; TaKaRa, Dalian, China), 1.0 μL each of 10 μM primer, 3.0 μL total genomic DNA approximately 100 ng as template and 18–22 μL of sterile distilled water. PCR was performed according to the reaction method as follows: 98 °C for 5 s, 30 cycles at 98 °C for 10 s, annealing at 56 °C for 30 s and a final extension at 72 °C for 2 min. Purified PCR products were directly sequenced using primers COIF and COIR from both ends in a semiautomated DNA analyzer (3700; Applied Biosystems, Waltham, MA, USA). To avoid the errors in amplification and sequencing, all singletons among polymorphism sites were verified by additional amplification and sequencing.

### 2.3. Genetic Diversity Analysis

COI sequences were edited and aligned using Clustal X 1.81(University College Dublin Belfield, Dublin, Ireland) software. Fish diversity was evaluated by determining COI haplotype diversity (Hd) and nucleotide diversity (π) [23]. These two indexes were calculated following Nei with DnaSP 5.01(Universitat de Barcelona, Barcelona, Spain) software [24]. Phylogenetic relationships among haplotypes were reconstructed using the neighbor-joining (NJ) method. We compared our COI sequences with *Protosalanx hyalocranius* (accession number MK523756.1) and genetic distances were generated for UPGMA phylogenetic tree reconstruction using models of substitution by MEGA 9.0 (Sudhir Kumar, Koichiro Tamura, and Masatoshi Neiand, The Pennsylvania State University, Philadelphia, PA, USA). Genetic differentiation was estimated by calculating pairwise distances between populations [25]. Pairwise genetic divergences between populations were estimated using the fixation index (Fst) considering genetic distances. The significance of Fst was tested by means of 1000 permutations of each pairwise comparison. This analysis and a hierarchical analysis of molecular variance (AMOVA) were carried out using Arlequin 3.0 (Laurent Excoffier, Bern, Switzerland) software to assess the partition variance components attributable to population variance and to individuals within populations [26]. The relationships among samples were visualized using a haplotype network constructed with Network 10.2(Fluxus Technology Ltd., London, UK) software [27]. Furthermore, the historical population expansion was examined using a neutrality test and the mismatch distribution in DnaSP v6.0. (Universitat de Barcelona, Barcelona, Spain) For the neutral test, the values of Tajima’s D and Fu’s FS were calculated [28,29]. The timing of possible population expansions (t) was calculated from the relationship t = Tτ /2 µk, where T was the generation time, the tau value (τ) was the mode of the mismatch distribution, µ was the mutation rate per nucleotide and k was the number of nucleotides of the analyzed fragment [30]. The time of initial maturity for *N.*
*taihuensis* was estimated to be one year [31]. We used the mutation rate range of mitochondrial coding genes per nucleotide per generation per million years [32] and the value of τ was calculated by Arlequin 3.5 [33].

## 3. Results

### 3.1. Genetic Diversity

All the COI sequences of mtDNA regions were homologous in length and aligned using cluster X. A total of 197 COI sequences with an aligned length of 647 bp were generated from six river basins. Eleven polymorphic sites and thirteen haplotypes were detected among all samples. The average nucleotide composition for all individuals was highest for G (33.32%), followed by A (26.04%), T (21.43%), and C (19.22%), The G/C contents (52.54%) were significantly higher than the A/T contents (47.47%), showing a strong G/C bias, respectively (Table 1).

In this study, 13 different haplotypes were identified in the 197 samples (Table 1 and Table A1). The number of haplotypes ranged from two to eight in each sampled basin. The FX population possessed the highest haplotype diversity compared with the other sampled populations. The lowest number of haplotypes was found in the YMQ population, in the Nenjiang River Basin. The haplotype frequency distribution was strongly skewed; the most common haplotype H4 was found in nine sampled populations, occupying 34.01% of all individuals among five river basins. Other shared common haplotypes found in all five basins were haplotype H1, H2 and H3, with 29.95%, 19.29% and 9.64% of all individuals, respectively (Table 2 and Table A2). H4 was found in nine populations and is likely to be an ancestral haplotype. Similarly, the nucleotide diversity ranged from 0.0006 (NjRB) to 0.0034 (YeRB), with the highest value in the LH population.

### 3.2. Population Differentiation

The pairwise comparisons Fst test showed that there were genetic differences among all 11 populations (Fst = −0.0916~0.5387, *p* < 0.05). The highest degree of genetic differentiation occurred between the TH and XLD populations (Fst = 0.5387), and the lowest degree of genetic differentiation occurred between the LH and LHP populations (Fst = −0.0916, Table 3). The UPGMA phylogenetic tree was constructed based on the genetic distance between the 11 populations. The genetic distance ranged from 0.0014 to 0.0031, and there was no significant geographical population pattern between populations. TH and XLD were the most distant (0.0031), while YMQ and XLD were the shortest (0.0014). LH was clustered into a single branch (Figure 2).

### 3.3. Population Structure

The haplotype network tree showed a star-like topology characterized by a large number of unique haplotypes radiating from a few dominant haplotypes. Each haplotype was connected with others by one to three mutational steps, which suggested that there was no deep branch among haplotypes for this species. In the network tree, H4 yielded the highest outgroup weight (0.3401) followed by H1 (0.2995), H2 (0.1929) and H3 (0.0964), which indicated H4 was the most probable ancestral haplotype, as it was shared with five river basins. In fact, a high frequency of shared haplotypes (H4, H1, H2, H3) was found among all basins, which produced a pattern of population relationships associated with the ancestry of the haplotypes found within each population rather than with the hydrological network (Figure 3).

The NJ (neighbor-joining) tree was based on the COI marker, and the same samples gave similar results (Figure 4). The NJ tree showed that haplotypes were mainly divided into three branches, and the *Protosalanx hyalocranius* served as a separate branch of the out group. The results of the NJ tree basically corresponded to the results of the haplotype network tree, but there was no significant population pattern, and haplotypes were cross-distributed among populations.

### 3.4. Historical Demography

Neutral test analysis showed that Fu*’*s Fs value and Tajima*’*s D test of the COI gene had negative values, but the statistical test was not significant (Tajima’s D = −0.226, *p* > 0.10; Fs = −2.037, *p* = 0.059 > 0.05), indicating that the population deviated from neutral evolution. The mismatch distribution showed a unimodal pattern (Figure 5). Both the neutral test and the mismatch distribution suggested that the *N. taihuensis* had experienced a population expansion in the past. In terms of the beginning of population expansion, t = Tτ/2µk was used to calculate the diffusion time of the *N. taihuensis* populations, the τ value of 1.053 derived based on COI suggested a time of approximately 0.041–0.051 MYA (million years ago) occurring during the Late Pleistocene.

## 4. Discussion

### 4.1. Genetic Diversity

Genetic studies are of great importance, since they can improve our understanding of genetic diversity and evolution, and therefore provide constructive guidance for the conservation and management of genetic resources, as well as the development of breeding programs [34,35]. Genetic diversity is to some extent the basis for the evolution and differentiation of animal species, which determines the ability of species to adapt to environmental changes [36]. In this study, the genetic diversity of 11 populations of 197 samples was analyzed by the molecular marker of the mitochondrial COI gene, and a total of 13 different haplotypes were identified. H4 was found within nine populations and may be considered the ancestral haplotype; the genetic diversity parameters were based on the proposed standard [37]. The overall haplotype diversity was greater than 0.5 (0.719 ± 0.145), and the nucleotide diversity was less than 0.005 (0.0021 ± 0.0007), suggesting that *N. taihuensis* may have experienced a rapid expansion of the population after the bottleneck effect cause [37]. Similarly, previous studies have recorded the haplotype and nucleotide diversity levels of mitochondrial DNA COI gene in the *N. taihuensis* population. Contrary to Gaoyou Lake (0.323 ± 0.085 and 0.00038 ± 0.00014, respectively) and Luoma Lake (0.361 ± 0.103 and 0.00062 ± 0.00062) [38], Dongting Lake (0.730 and 0.0013) and Chaohu Lake (0.667 ± 0.113 and 0.00160 ± 0.0191) showed a high haplotype diversity and low nucleotide diversity [39].

Changes in fish genetic diversity are affected by many factors, including geographic isolation, human activities, habitat differences and mitochondrial evolution rates [40]. The extensive high haplotype diversity may be attributed to the complex and changeable distribution of *N. taihuensis* populations in China. The higher haplotype and lower nucleotide results are basically consistent with previous studies [41]. The results of this study showed that the haplotype diversity (0.880) of the FX population was the highest, and the nucleotide diversity (0.0006) of the XLD population was the lowest. The genetic diversity of the six watersheds was ranked as YeRB > LcRB > HhRB > YzRB > NjRB > YzRB, thus revealing that the genetic diversity in the YzRB was low, which was consistent with previous studies in *N. taihuensis* [42,43].

The genetic diversity and population structure of the same species in different geographical locations are often related to the living environment [44]. The Yangtze River Basin has a large area, many tributaries and connected lakes. It has better habitat conditions for fish growth and development. In contrast, the Yangtze River Basin should have more populations and a higher genetic diversity. However, the results of this study showed that the genetic diversity level of the YzRB population was lower than that of the other five river basins, which may be due to overfishing, dam construction and water pollution in the Yangtze River Basin, resulting in a sharp decline in wild population resources; for *N. taihuensis* as a usually one-year-lived “R-strategy” creature, a high fecundity and abundant resources are important biological competition strategies. In addition, combined with its inherent biological characteristics, *N. taihuensis* is a cold-water fish, and the suitable water temperature is low [45]. The latitudes of the YeRB and ShRB are higher, and the water temperature is lower than that of the YzRB [45]. The LcRB belongs to the Pacific water system and originates from a small glacier at an altitude of 5167 m, which is suitable for the growth and development of *N. taihuensis*.

### 4.2. Genetic Differentiation

In this study, the COI gene sequence was used to analyze genetic differentiation among the populations of *N. taihuensis*. An AMOVA showed that the genetic variation of *N. taihuensis* mainly came from within the population (90.37%). The degree of differentiation was at a moderate level (0.05 ≤ Fst = 0.09 < 0.25), and among different basins, the genetic differentiation was the largest (Fst = 0.5387) between the TH and XLD populations. Within the same basin, the genetic differentiation (Fst = 0.2146) between the HSH and LHP populations was the highest, and the maximum variation rate between different basins was higher than that within the same basin. It may be that geographic isolation affects the genetic differentiation of species, resulting in different genetic diversity of species, but this study found that no obvious geographical pattern was formed, and the genetic differences between groups were low [35].

### 4.3. Population Structure

From the perspective of phylogenetic evolution, the analysis of the UPGMA tree based on genetic distance showed that their close relationship and geographic proximity were basically consistent. This study took population samples from different waters in the north and south, and the UPGMA tree was roughly divided into three branches (Clade 1, Clade 2 and Clade 3); the NjRB, Songhua River and XLD population in the northern water system formed Clade 2; the southern water system YzRB, LcRB and HhRB formed Clade 3. Due to the inherent biological characteristics of Clade 1, most samples were in relatively closed water bodies for a long time, and the gene exchange with other groups was limited, forming a distance isolation model organism and resulting in a long genetic distance. The haplotype network map constructed from the 13 haplotypes was radically distributed around the main haplotype, and different groups intersected with each other, which basically corresponded to the results of the NJ phylogenetic tree. There was no significant group pattern.

### 4.4. Historical Dynamics

In general, patterns of genetic diversity in species are closely related to their evolutionary history. Population historical evolution is usually detected in two ways: one is to observe the nucleotide unpaired distribution map based on the infinite-site model; the other is neutrality tests [30,46]. In this study, Tajima’s D-test and Fu’s Fs-test showed negative values, and significant differences in *p* values can be considered as showing historical signs of population expansion [28,47,48]. Moreover, the base unpairing distribution curve had a unimodal shape, which revealed that *N. taihuensis* may have experienced population expansion. *N. taihuensis* originated in the middle of the Tertiary and experienced alternating Quaternary and interglacial climate during its evolution [48]. We estimated the expansion time of *N. taihuensis* population at 0.041–0.051 million years, both earlier than the last major glaciation of the Quaternary glacial period, about 18,000 years ago. It is speculated that the last large glacial period had a great impact on the *N. taihuensis* population, during which *N. taihuensis* lived in a shelter. With the end of the last glacial period, the warming caused the sea level rising, and the *N. taihuensis* population in Taihu Lake spread out [49]. During the subsequent evolution, different haplotypes gradually emerged, forming present-day patterns of genetic structure.

### 4.5. Research Prospective

As one of the most important commercial species in the salangid family, the fish *N. taihuensis* have been attracting significant attention in China. It is worth noticing that the population size of *N. taihuensis* and its fishing yield have decreased rapidly in recent years [50]. Overfishing, water pollution and habitat fragmentation were also proven to be the vital threat to this species [51]. Consequently, most of the small lakes nearly disappeared and the mid-large lakes shrank considerably resulting in unsuitable habitat for this species [52,53]. At the moment, no specific protection action has been taken for *N. taihuensis*, so it is urgent to make an effective plan to protect this economically and ecological important species. Furthermore, as a one-year-lived species, all populations should be managed and conserved separately [45]. Moreover, special attention must be paid to habitat protection, reproductive biology and a consistent artificial propagation and enhancement coming from *N. taihuensis*’s nature reservoir.

## 5. Conclusions

The results showed a high haplotype diversity and a low nucleotide diversity between the *N. taihuensis* populations, further indicating that the population had an overall genetic diversity The low level of genetic diversity may lead to a reduction of species fertility and an increase of harmful recessive factors, which result in species degradation. As a native fish species, *N. taihuensis* should be paid enough attention to maintain its genetic diversity. The YzRB served as the main origin for this species, and it should be protected more. As a usually one-year-lived “R-strategy” creature, the *N. taihuensis* is more sensitive to habitat changes, so habitat protection and water pollution control should be given priority. In a later stage, more nuclear marker methods can be analyzed to explore the genetic diversity, in order to provide more basic information for the germplasm resources of *N. taihuensis*.

## Figures and Tables

**Figure 1 biology-11-00968-f001:**
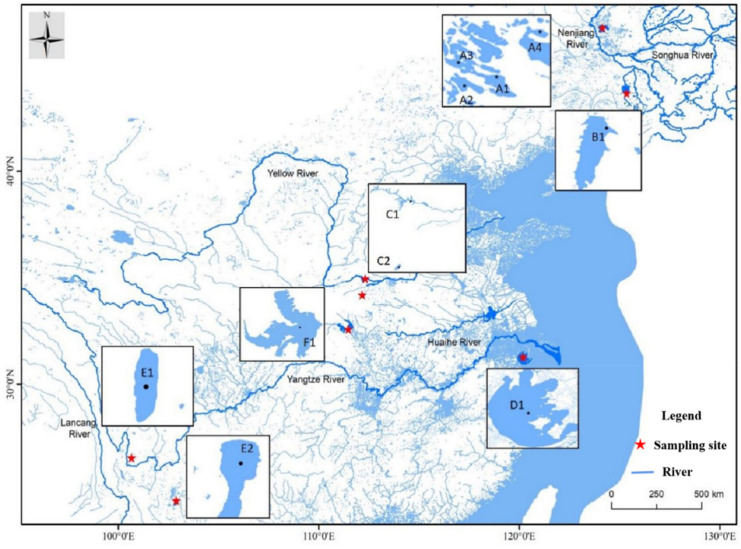
Map showing sampling locations of *N. taihuensis* in six different river basins in China. Note: NjRB: Nenjiang River Basin, A1 to A4 represents Huoshaohei lake (HSH), Amuta lake (AMT), Yamenqi lake (YMQ), Longhupao lake (LHP), respectively; ShRB: Songhua River Basin, B1: Xinlicheng reservoir (XLC); YeRB: Yellow River Basin, C1: Xiaolangdi reservoir (XLD), C2: Luhun reservoir (LH); YzRB: Yangtze River Basin, D1: Taihu lake (TH); LcRB: Lanchang River Basin, E1; Chenghai (CH), E2: Fuxian lake (FX) and HhRB: Huaihe River Basin, F1: Hongze lake (HZ).

**Figure 2 biology-11-00968-f002:**
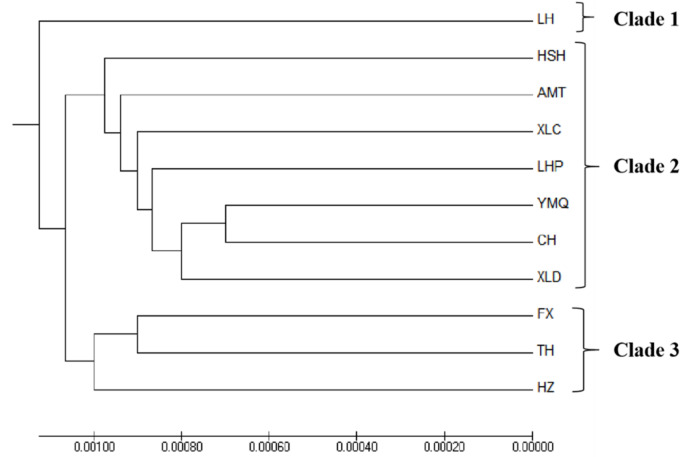
UPGMA phylogenetic trees of *N. taihuensis* from the 11 populations. The AMOVA results revealed that 90.37% of genetic variation occurred within populations, whereas 9.63% of genetic variation occurred among populations. An AMOVA of the 11 populations yielded an Fst value of 0.6452 (*p* < 0.01), which suggested significant genetic variation among the 11 populations (Table 4).

**Figure 3 biology-11-00968-f003:**
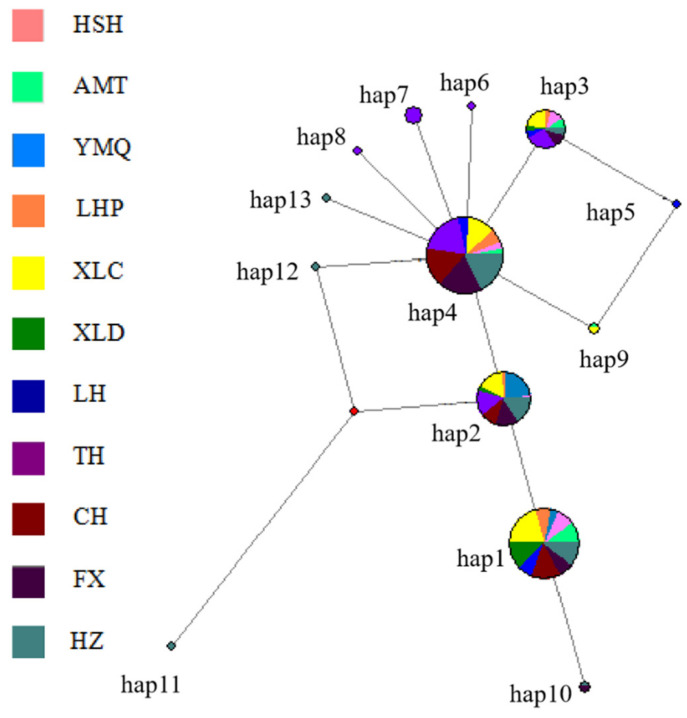
Haplotype network trees of *N. taihuensis* in different populations.

**Figure 4 biology-11-00968-f004:**
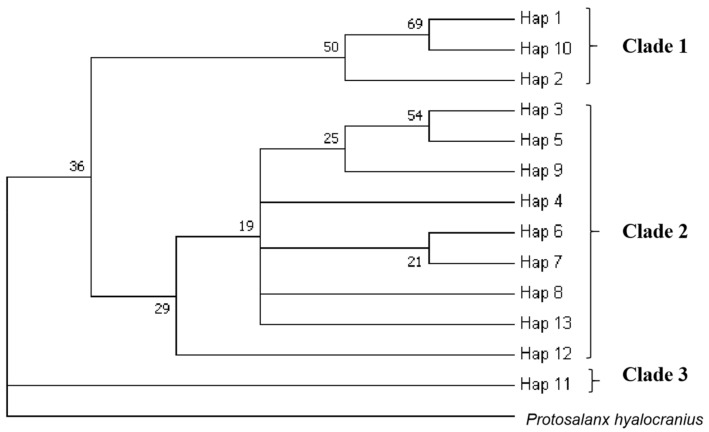
NJ tree of *N. taihuensis* in different populations by the Kimura 2-parameter model. Note: Numbers at nodes represent bootstrap values.

**Figure 5 biology-11-00968-f005:**
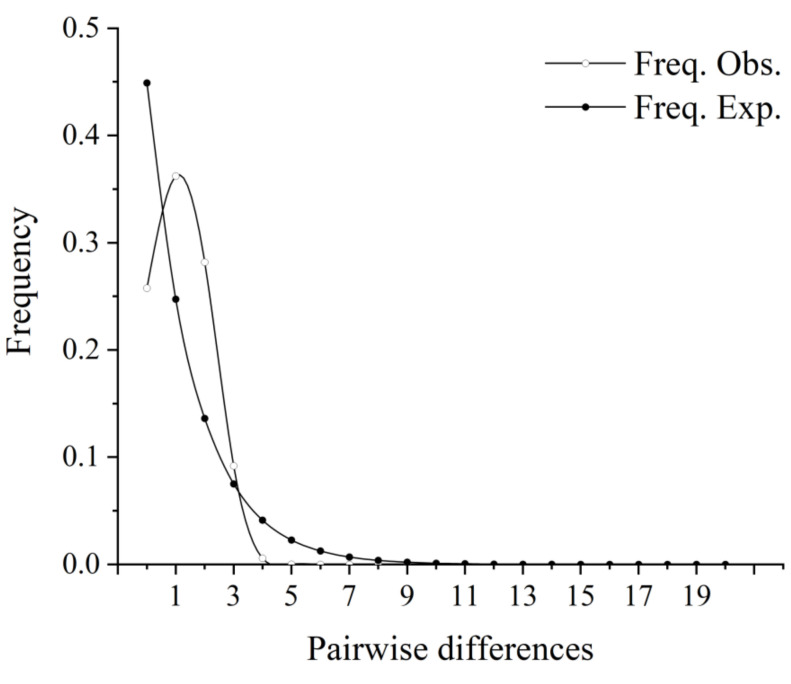
The observed pairwise difference and the expected mismatch distributions of *N. taihuensis*.

**Table 1 biology-11-00968-t001:** The average nucleotide composition of *N. taihuensis* in different populations.

Populations Codes	T (%)	C (%)	A (%)	G (%)	T + A (%)	C+G (%)
Nenjiang River Basin						
A1 (Huoshaohei lake, HSH)	21.45	19.20	26.15	33.20	47.60	52.40
A2 (Amuta lake, AMT)	21.45	19.20	26.13	33.21	47.58	52.41
A3 (Yamenqi lake, YMQ)	21.48	19.17	26.15	33.19	47.63	52.36
A4 (Longhupao lake, LHP)	21.47	19.19	26.11	33.25	47.58	52.44
Songhua River Basin						
B1 (Xinlicheng reservoir, XLC)	21.46	19.20	26.12	33.24	47.58	52.44
Yellow River Basin						
C1 (Xiaolangdi reservoir, XLD)	21.47	19.18	26.21	33.14	47.68	52.32
C2 (Luhun reservoir, LH)	21.43	19.22	26.05	33.30	47.48	52.52
Yangtze River Basin						
D1 (Taihu lake, TH)	21.44	19.21	25.99	33.37	47.43	52.58
Lanchang River Basin						
E1 (Chenghai, CH)	21.36	19.30	25.95	33.41	47.31	52.71
E2 (Fuxian lake, FX)	21.39	19.27	25.92	33.42	47.31	52.69
Huaihe River Basin						
F1 (Hongze lake, HZ)	21.42	19.20	25.99	33.40	47.41	52.60
Total average	21.43	19.22	26.04	33.32	47.47	52.54

**Table 2 biology-11-00968-t002:** Genetic diversity indices of different *N. taihuensis* populations.

Population	H1	H2	H3	H4	H5	H6	H7	H8	H9	H10	H11	H12	H13	N	Hd	s	k	π
Nenjiang River Basin														40	0.724	4	1.186	0.0018
A1 (Huoshaohei lake)	6		1	2					1					10	0.644	4	1.467	0.0022
A2 (Amuta lake)	5	1	2	2										10	0.733	3	1.444	0.0022
A3 (Yamenqi lake)	2	8												10	0.356	1	0.356	0.0006
A4 (Longhupao lake)	4	1	1	4										10	0.733	3	1.289	0.0020
Songhua River Basin														31	0.753	4	1.290	0.0020
B1 (Xinlicheng reservoir)	12	6	4	8					1					31	0.753	4	1.290	0.0020
Yellow River Basin														19	0.754	5	1.684	0.0026
C1 (Xiaolangdi reservoir)	8	1	1											10	0.533	4	0.956	0.0015
C2 (Luhun reservoir)	4		1	3	1									9	0.861	5	2.222	0.0034
Yangtze River Basin														30	0.789	6	1.120	0.0017
D1 (Taihu lake)		6	5	13		1	4	1						30	0.789	6	1.120	0.0017
Lanchang River Basin														47	0.826	6	1.576	0.0024
E1 (Chenghai)	8	4		11										23	0.771	3	1.296	0.0020
E2 (Fuxian lake)	4	5	2	12						1				24	0.880	6	1.819	0.0028
Huaihe River Basin														30	0.777	7	1.407	0.0022
F1 (Hongze lake)	6	6	2	12						1	1	1	1	30	0.777	7	1.407	0.0022
Entire region (all samples)	59	38	19	67	1	1	4	1	2	2	1	1	1	197	0.724	9	1.237	0.0022

Note: N, number of individuals; Hd, haplotype diversity; S, number of segregating sites; k, mean pairwise nucleotide; π, nucleotide diversity.

**Table 3 biology-11-00968-t003:** The F_ST_ value (below diagonal) and genetic distance (above diagonal) among 11 populations of *N. taihuensis* based on COI sequence data.

Populations	NjRB				ShRB	YeRB		YzRB	LcRB		HhRB
	A1 (HSH)	A2 (AMT)	A3 (YMQ)	A4 (LHP)	B1 (XLC)	C1 (XLD)	C2 (LH)	D1 (TH)	E1 (CH)	E2 (FX)	F1 (HZ)
Nenjiang River Basin											
A1 (HSH)		0.0021	0.0018	0.0020	0.0021	0.0018	0.0023	0.0026	0.0019	0.0022	0.0024
A2 (AMT)	−0.0862		0.0017	0.0020	0.0020	0.0018	0.0023	0.0024	0.0020	0.0020	0.0023
A3 (YMQ)	0.2146 **	0.1818 **		0.0016	0.0016	0.0014	0.0021	0.0021	0.0014	0.0016	0.0018
A4 (LHP)	−0.0598	−0.0847	0.2094 **		0.0019	0.0019	0.0022	0.0021	0.0017	0.0018	0.0020
Songhua River Basin											
B1 (XLC)	−0.0285	−0.0545	0.1278 *	−0.0633		0.0019	0.0022	0.0022	0.0018	0.0019	0.0021
C1 (XLD)	0.0253	0.0678 *	0.3687 **	0.1756 **	0.1446 *		0.0023	0.0031	0.0018	0.0023	0.0024
C2 (LH)	−0.0702	−0.0871	0.2454 **	−0.0916	−0.0472	0.1617 **		0.0024	0.0021	0.0022	0.0024
Yangtze River Basin											
D1 (TH)	0.3035 **	0.2458 **	0.4443 **	0.1610 **	0.1966 **	0.5387 **	0.1544 **		0.0019	0.0018	0.0021
Lanchang River Basin											
E1 (CH)	0.0074	−0.0103	0.1920 **	−0.0633	−0.0187	0.2391 *	−0.0127	0.1908 **		0.0016	0.0019
E2 (FX)	0.0869 *	0.0399	0.2441 **	−0.0321	0.0157	0.3401 **	−0.0049	0.0718 *	−0.0052		0.0019
Huaihe River Basin											
F1 (HZ)	0.0524 *	0.0155	0.1614 **	−0.0391	0.0054	0.2584 **	−0.0102	0.0948 *	−0.0137	−0.0286	

Note: Asterisks * indicate significant values after Bonferroni correction. * (0.01 < *p* < 0.05), ** (*p* < 0.01).

**Table 4 biology-11-00968-t004:** AMOVA of *N. taihuensis* populations.

Source of Variation	d. f.	Sum of Squares	Variance Components	Percentage of Variation
Among population	10	16.673	0.06215Va	9.63
Within population	186	108.450	0.58307Vb	90.37
Total	196	125.122	0.6452	
Fixation		Fst	0.6452	

## Data Availability

Not applicable.

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
