# Peer review of "Assessment of Genetic Diversity of the Salangid, Neosalanx taihuensis, Based on the Mitochondrial COI Gene in Different Chinese River Basins"

_biology, 2022, doi:10.3390/biology11070968_

Round 1
Reviewer 1 Report
The authors of study entitled “Assessment of genetic diversity of the Salangid, Neosalanx taihuensis, based on the mitochondrial COI gene in Chinese different river basins” aimed to evaluate the genetic diversity and population structure of 197 individuals of Neosalanx taihuensis belonging to 11 populations from six Chinese rivers basins, by analysing the COI gene sequence. The paper is quite well written in general. However, there are some points that must be better explained and deepened.
Major comments:
I think authors should add a paragraph and some references concerning the analysis of COI gene sequence in fish. It completely lacks in the introduction section.
Figure 1: do stars indicate the location of each basin? Please make it explicit in figure legend.
Just a note: when writing a manuscript, numbers from one to ten should be written in letters.
To better describe the results and allow readers to follow the discussion, I think authors should add a table (also as supplementary) reporting different information for each COI haplotype, such as mutations, frequency in each population and basin, frequency among all samples, …
Figure 3: add “Refer to Table 1 for the population codes”. I also think you could remove the underscore in the haplotype codes.
The manuscript completely lacks a comparison with other populations of Neosalanx taihuensis, also in term of COI haplotype diversity. Is there no data that authors could refer to?
Minor comments:
Line 12: would you replace “from” with “of”?
Line 18: replace “suggested” with “suggest that”.
Line 18-19 and 37: what do you mean with “management units”? Please explain.
Lines 21: delete “which is”.
Lines 27: replace “were” with “was”.
Line 28: what do you mean with “Haplotype and its frequency distributions”? Why did you use “haplotype” in the singular form?
Line 28: replace “hapltypes” with “haplotypes”.
Line 30: replace “among” with “in”.
Line 30: replace “6” with “six”.
Line 36: modify “N. taihuensis populations experienced population expansion…” with “N. taihuensis experienced an expansion…”.
Line 44: replace “Population genetic structure” with “The genetic structure of a population”.
Line 45: replace “Different basin populations” with “Populations from different basins”.
Line 51: add a comma after “taihuensis”.
Line 52: “which is”.
Line 55: delete “was”.
Line 58: delete “have”.
Line 59: “for the assessment”.
Line 60-61: replace “nuclear DNA markers and mitochondrial DNA (MtDNA) markers” with “nuclear and mitochondrial (mtDNA) DNA markers”.
Line 64: “it is maternally inherited”.
Line 65: replace “its” with “the”.
Line 68-73: re-write the sentence as follows “In this regard, the COI gene sequence from 11 populations was analysed in order to detect the population structure and the genetic diversity of N. taihuensis at large geographical scales. Samples were collected from Nenjiang River Basin (NjRB, A1-A4), Songhua River Basin (ShRB, B1), Yellow River Basin (YeRB, C1-C2), Yangtze River Basin (YzRB, D1), Lanchang River Basin (LcRB, E1-E2) and Huaihe River Basin (HhRB, F1)”.
Line 76: “A total of”.
Line 113 and 195: “Protosalanx hyalocranius” in cursive.
Line 115: please, correct with “pairwise” throughout the manuscript.
Line 121: “among samples”.
Line 136: replace “6” with “six”.
Line 141: replace “individuals” with “populations”.
Line 144: “from two to eight”.
Line 146-147: re-write the sentence as follows “The lowest number of haplotypes was found in YMQ population, in Nenjiang River Basin”.
Line 148: “nine”.
Line 149: “five”.
Line 151: “H4 was found in nine populations”.
Line 173: add spaces between “11” and “populations” and after the dot.
Line 175: delete “the” before “genetic”.
Line 186: “five river basins”.
Line 187: “was found”.
Line 193: “The NJ (Neighbor-Joining) tree…”.
Line 200: add a space before “Note”.
Line 207: “experienced a population…”.
Line 226-227: “H4 was found within nine populations and maybe considered the ancestral haplotype”.
Line 228: replace “The haplotype diversity of all was…” with “The overall haplotype diversity was…”.
Line 230: “experienced a rapid”.
Line 234: “in my country”?
Line 234: replace “This” with “The”.
Line 239: replace “NjRB > YzRB. The results revealed” with “NjRB > YzRB, thus revealing”.
Line 247: “five”.
Line 254: “5,167”.
Line 275: in some cases, the word “clade” is written with capital letter and without a space before the number, please make it uniform throughout the manuscript.
Line 278: “…organism and resulting…”.
Line 288: “[18,32]. Moreover”.
Line 292-293: what do you mean with “historically”? Please, re-write the sentence.
Line 297: “population” without capital letter.
Line 299: replace “to rise” with “rising”.
Line 306: add a dot before “Over-fishing”.
Line 317: “six”.
Line 317: replace “that there was” with “a”.
Line 318: “and a low”.
Line 318-319: “populations, further indicating that the population had an overall genetic diversity”.
Line 323: “species and it”.
Line 326: “more nuclear marker methods can be analysed to explore the genetic diversity”.
Author Response
Response to the Review 1 Comments
We sincerely thank for your careful reading, helpful comments, and constructive suggestions, which has significantly improved the presentation of our manuscript. We have tried our best to revise the manuscript and provided a point-by-point response to the reviewers' comments below in blue color.
Response to the Review 1
The authors of study entitled “Assessment of genetic diversity of the Salangid, Neosalanx taihuensis, based on the mitochondrial COI gene in Chinese different river basins” aimed to evaluate the genetic diversity and population structure of 197 individuals of Neosalanx taihuensis belonging to 11 populations from six Chinese rivers basins, by analysing the COI gene sequence. The paper is quite well written in general. However, there are some points that must be better explained and deepened.
Response: We gratefully thanks for the precious time the reviewer spent making constructive remarks.
Major comments:
1.I think authors should add a paragraph and some references concerning the analysis of COI gene sequence in fish. It completely lacks in the introduction section.
Response: Thanks for your kind suggestions. In introduction section, more descriptions about the analysis of COI gene sequence in fish have been included in the revised manuscript.
2.Figure 1: do stars indicate the location of each basin? Please make it explicit in figure legend.
Response: The star indicates the location of each basin, and the legend of sampling sites and river is added in the manuscript.
- Just a note: when writing a manuscript, numbers from one to ten should be written in letters.
Response: Thank you very much for your reminding. we have made changes in the article.
4.To better describe the results and allow readers to follow the discussion, I think authors should add a table (also as supplementary) reporting different information for each COI haplotype, such as mutations, frequency in each population and basin, frequency among all samples, …
Response: We appreciate the insightful suggestion. We have added two tables in the appendix of the article to increase the understanding of haplotype characteristics and the distribution frequency of haplotypes in each population and basin.
Appendix A Variable sites of mitochondrial DNA COI gene haplotypes in N.taihuensis
|
Haplotype |
Variable sites |
||||||||||
|
35 |
75 |
303 |
315 |
393 |
453 |
483 |
507 |
645 |
646 |
647 |
|
|
Hap1 |
C |
T |
A |
A |
A |
G |
T |
T |
T |
C |
C |
|
Hap2 |
C |
T |
G |
A |
A |
G |
T |
T |
T |
C |
C |
|
Hap3 |
C |
T |
G |
A |
A |
G |
T |
C |
T |
C |
C |
|
Hap4 |
C |
T |
G |
A |
A |
G |
T |
T |
T |
C |
C |
|
Hap5 |
C |
T |
G |
G |
A |
G |
C |
C |
T |
C |
C |
|
Hap6 |
C |
T |
G |
A |
G |
G |
T |
T |
T |
C |
C |
|
Hap7 |
C |
C |
G |
A |
A |
G |
T |
T |
T |
C |
C |
|
Hap8 |
C |
T |
G |
A |
A |
A |
A |
T |
T |
C |
C |
|
Hap9 |
C |
T |
G |
A |
A |
G |
T |
T |
T |
C |
C |
|
Hap10 |
T |
T |
A |
A |
A |
G |
T |
T |
T |
C |
C |
|
Hap11 |
C |
T |
A |
A |
A |
G |
T |
T |
C |
G |
A |
|
Hap12 |
C |
T |
G |
A |
A |
G |
T |
T |
C |
C |
C |
|
Hap13 |
C |
T |
G |
A |
A |
G |
T |
T |
T |
C |
T |
Appendix B Frequency in each population and basin of COI gene haplotypes in N.taihuensis
|
Haplotype |
Hap1 |
Hap2 |
Hap3 |
Hap4 |
Hap5 |
Hap6 |
Hap7 |
Hap8 |
Hap9 |
Hap10 |
Hap11 |
Hap12 |
Hap13 |
|
NjRB |
0.2881 |
0.2368 |
0.2105 |
0.1194 |
0.5 |
||||||||
|
ShRB |
0.2034 |
0.1316 |
0.2105 |
0.1194 |
0.5 |
||||||||
|
YeRB |
0.2034 |
0.0263 |
0.1053 |
0.0448 |
1 |
||||||||
|
YzRB |
0.1579 |
0.2632 |
0.1940 |
1 |
1 |
1 |
|||||||
|
LcRB |
0.2034 |
0.2368 |
0.1053 |
0.3433 |
0.5 |
||||||||
|
HhRB |
0.1017 |
0.1579 |
0.1053 |
0.1791 |
0.5 |
1 |
1 |
1 |
|||||
|
HSH |
0.1017 |
0.0526 |
0.0299 |
0.5 |
|||||||||
|
AMT |
0.0847 |
0.0263 |
0.1053 |
0.0299 |
|||||||||
|
YMQ |
0.0339 |
0.2105 |
|||||||||||
|
LHP |
0.0678 |
0.0263 |
0.0526 |
0.0597 |
|||||||||
|
XLC |
0.2034 |
0.1579 |
0.2105 |
0.1194 |
0.5 |
||||||||
|
XLD |
0.1356 |
0.0263 |
0.0526 |
||||||||||
|
LH |
0.0678 |
0.0526 |
0.0448 |
1 |
|||||||||
|
TH |
0.1579 |
0.2632 |
0.1940 |
1 |
1 |
1 |
|||||||
|
CH |
0.1356 |
0.1053 |
0.1642 |
||||||||||
|
FX |
0.0678 |
0.1316 |
0.1053 |
0.1791 |
0.5 |
||||||||
|
HZ |
0.1017 |
0.1579 |
0.1053 |
0.1791 |
0.5 |
1 |
1 |
1 |
5.Figure 3: add “Refer to Table 1 for the population codes”. I also think you could remove the underscore in the haplotype codes.
Response:We really appreciate your suggestions. We have modified table 1 to improve the naming of haplotypes in Figure 4.
Figure 3 Haplotype network trees of N.taihuensis in different populations
6.The manuscript completely lacks a comparison with other populations of Neosalanx taihuensis, also in term of COI haplotype diversity. Is there no data that authors could refer to?
Response: We appreciate the insightful suggestion. In discussion section, In the discussion section, we added the content of coI haplotype diversity and compared the results of this study with those of previous studies, including the levels of haplotype diversity and nucleotide diversity. more descriptions have been included in the revised manuscript.
Minor comments:
Line 12: would you replace “from” with “of”?
Line 18: replace “suggested” with “suggest that”.
Line 18-19 and 37: what do you mean with “management units”? Please explain.
Lines 21: delete “which is”.
Lines 27: replace “were” with “was”.
Line 28: what do you mean with “Haplotype and its frequency distributions”? Why did you use “haplotype” in the singular form?
Line 28: replace “hapltypes” with “haplotypes”.
Line 30: replace “among” with “in”.
Line 30: replace “6” with “six”.
Line 36: modify “N. taihuensis populations experienced population expansion…” with “N. taihuensis experienced an expansion…”.
Line 44: replace “Population genetic structure” with “The genetic structure of a population”.
Line 45: replace “Different basin populations” with “Populations from different basins”.
Line 51: add a comma after “taihuensis”.
Line 52: “which is”.
Line 55: delete “was”.
Line 58: delete “have”.
Line 59: “for the assessment”.
Line 60-61: replace “nuclear DNA markers and mitochondrial DNA (MtDNA) markers” with “nuclear and mitochondrial (mtDNA) DNA markers”.
Line 64: “it is maternally inherited”.
Line 65: replace “its” with “the”.
Line 68-73: re-write the sentence as follows “In this regard, the COI gene sequence from 11 populations was analysed in order to detect the population structure and the genetic diversity of N. taihuensis at large geographical scales. Samples were collected from Nenjiang River Basin (NjRB, A1-A4), Songhua River Basin (ShRB, B1), Yellow River Basin (YeRB, C1-C2), Yangtze River Basin (YzRB, D1), Lanchang River Basin (LcRB, E1-E2) and Huaihe River Basin (HhRB, F1)”.
Line 76: “A total of”.
Line 113 and 195: “Protosalanx hyalocranius” in cursive.
Line 115: please, correct with “pairwise” throughout the manuscript.
Line 121: “among samples”.
Line 136: replace “6” with “six”.
Line 141: replace “individuals” with “populations”.
Line 144: “from two to eight”.
Line 146-147: re-write the sentence as follows “The lowest number of haplotypes was found in YMQ population, in Nenjiang River Basin”.
Line 148: “nine”.
Line 149: “five”.
Line 151: “H4 was found in nine populations”.
Line 173: add spaces between “11” and “populations” and after the dot.
Line 175: delete “the” before “genetic”.
Line 186: “five river basins”.
Line 187: “was found”.
Line 193: “The NJ (Neighbor-Joining) tree…”.
Line 200: add a space before “Note”.
Line 207: “experienced a population…”.
Line 226-227: “H4 was found within nine populations and maybe considered the ancestral haplotype”.
Line 228: replace “The haplotype diversity of all was…” with “The overall haplotype diversity was…”.
Line 230: “experienced a rapid”.
Line 234: “in my country”?
Line 234: replace “This” with “The”.
Line 239: replace “NjRB > YzRB. The results revealed” with “NjRB > YzRB, thus revealing”.
Line 247: “five”.
Line 254: “5,167”.
Line 275: in some cases, the word “clade” is written with capital letter and without a space before the number, please make it uniform throughout the manuscript.
Line 278: “…organism and resulting…”.
Line 288: “[18,32]. Moreover”.
Line 292-293: what do you mean with “historically”? Please, re-write the sentence.
Line 297: “population” without capital letter.
Line 299: replace “to rise” with “rising”.
Line 306: add a dot before “Over-fishing”.
Line 317: “six”.
Line 317: replace “that there was” with “a”.
Line 318: “and a low”.
Line 318-319: “populations, further indicating that the population had an overall genetic diversity”.
Line 323: “species and it”.
Line 326: “more nuclear marker methods can be analysed to explore the genetic diversity”.
Response: We are very grateful to you for your suggestions. This part of the comments has been directly revised in the article as required and marked with a yellow background.

Reviewer 2 Report
The study is well designed and the entire manuscript is well written. I definitely recommend this paper for publication in Biology with some minor changes. Please see the attached file for my specific comments.

Author Response
Dear reviewer,
We sincerely thank for your careful reading, helpful comments, and constructive suggestions, which has significantly improved the presentation of our manuscript. We have tried our best to revise the manuscript and provided a point-by-point revise to the reviewers' comments in yellow color.
Reviewer 3 Report
Review
General comments: The manuscript entitled “Assessment of genetic diversity of the Salangid,
Neosalanx taihuensis, based on the mitochondrial COI gene in Chinese different river basins”
presents important analises assessing the genetic diversity of 197 samples from 11
populations in 6 river basins. The manuscript is well developed, presented, the M&M are in
accordance with the obtained results and the discussion is pertinente. Before the final
acceptance of the manuscript we recommend a few changes as listed bellow.
Introduction:
According to the International Rule of Zoological Nomenclature the first time a species is cited
in the text it should be mentioned in full and followed by its Author and date
Geographic location of the river basins in the introduction must be improved. There is a map in the M&M
however, we recommend some descriptive information in the introduction about the river basins
Legends must be self explanatory. All of them must be revised. We recommend revising all of them, using the full name of the species and when necessary including more detailed
information. Sepecially in the legend of figs 1- (please include the name of the country, the
region of the river basins studied); fig 4 (please include the name of the studied species) and so on
Discussion
Pg 4 line 234 I recommend changing “in my country” for “in China
Line 272 instead of mentioning “from north and South,” I would suggest including the distance between the most distant points; even approximately
Line 291 Paragraph is too long
LIne 297 Please correct “N. taihuensis Population “ for N. taihuensis population
Line 311 Is it possible to include a reference for this statement “Furthermore, as a kind of one-year-lived species, all populations should be managed and conserved separately”
Conclusion
Please give brief details on the river basin studied (LInes 316 e 317).
In the sentence “As a usually one-year-lived &R-strategy" creature” I suggest changing “a” for
“an”. English is not my first language, therefore it might be important to carry out an English language review.
Please check if the verb “resulted” is correct, in the line 321
References must be revised specially the scientific names.
Author Response
Dear Review,
We sincerely thank for your careful reading, helpful comments, and constructive suggestions, which has significantly improved the presentation of our manuscript. We have tried our best to revise the manuscript and provided a point-by-point response to the reviewers' comments below in yellow color.
